# Removal of As(III) from Water Using the Adsorptive and Photocatalytic Properties of Humic Acid-Coated Magnetite Nanoparticles

**DOI:** 10.3390/nano10081604

**Published:** 2020-08-15

**Authors:** Phuong Pham, Mamun Rashid, Yong Cai, Masafumi Yoshinaga, Dionysios D. Dionysiou, Kevin O’Shea

**Affiliations:** 1Department of Chemistry and Biochemistry, Florida International University, Miami, 11200 SW 8th ST, Miami, FL 33199, USA; ppham006@fiu.edu (P.P.); mmrashid@fiu.edu (M.R.); cai@fiu.edu (Y.C.); 2Southwest Environmental Research Center, Florida International University, 11200 SW 8th ST, Miami, FL 33199, USA; 3Department of Cellular Biology and Pharmacology, Herbert Wertheim College of Medicine, Florida International University, Miami, FL 33199, USA; myoshina@fiu.edu; 4Environmental Engineering and Science Program, Department of Chemical and Environmental Engineering, University of Cincinnati, Cincinnati, OH 45221, USA; dionysdd@ucmail.uc.edu

**Keywords:** humic acid-coated magnetite nanoparticles, photocatalysis, arsenic

## Abstract

The oxidation of highly toxic arsenite (As(III)) was studied using humic acid-coated magnetite nanoparticles (HA-MNP) as a photosensitizer. Detailed characterization of the HA-MNP was carried out before and after the photoinduced treatment of As(III) species. Upon irradiation of HA-MNP with 350 nm light, a portion of the As(III) species was oxidized to arsenate (As(V)) and was nearly quantitatively removed from the aqueous solution. The separation of As(III) from the aqueous solution is primarily driven by the strong adsorption of As(III) onto the HA-MNP. As(III) removals of 40–90% were achieved within 60 min depending on the amount of HA-MNP. The generation of reactive oxygen species (•OH and ^1^O_2_) and the triplet excited state of HA-MNP (^3^HA-MNP*) was monitored and quantified during HA-MNP photolysis. The results indicate ^3^HA-MNP* and/or singlet oxygen (^1^O_2_) depending on the reaction conditions are responsible for converting As(III) to less toxic As(V). The formation of ^3^HA-MNP* was quantified using the electron transfer probe 2,4,6-trimethylphenol (TMP). The formation rate of ^3^HA-MNP* was 8.0 ± 0.6 × 10^−9^ M s^−1^ at the TMP concentration of 50 µM and HA-MNP concentration of 1.0 g L^−1^. The easy preparation, capacity for triplet excited state and singlet oxygen production, and magnetic separation suggest HA-MNP has potential to be a photosensitizer for the remediation of arsenic (As) and other pollutants susceptible to advanced oxidation.

## 1. Introduction

Arsenic is widely distributed in the environment through natural and anthropogenic sources [1,2,3]. Arsenic can undergo reactions of oxidation–reduction, precipitation–dissolution, adsorption–desorption, as well as organic and biochemical methylation depending on environmental and biological conditions [2,4]. Acute and chronic toxicities of arsenic species depend on the valence state, speciation, availability, and animal species or cell types exposed to arsenic [2,5]. The trivalent arsenic species (As(III)) is considerably more toxic than the pentavalent species (As(V)), due to the high binding affinity for thiol ligands of biomolecules such as glutathione, lipoic acid, and cysteine residues of enzymes or proteins [5,6,7,8]. The bioavailability of As is primarily dictated by the speciation [1], with As(III) being more mobile and generally exhibiting lower adsorption affinity than As(V) [9]. Oxidation of As(III) to As(V) is desirable and often required for the efficient remediation of As species. 

Humic acid (HA), a fraction of natural organic matter (NOM), plays an important role in ecosystems. HA is a heterogeneous mixture of polycationic and anionic oligomer/polymer-based materials produced largely from terrestrial and microbial origins. HA can affect the transport and bioavailability of organic and inorganic pollutants in aquatic environments [10]. A number of HA constituents can strongly complex metal ions including arsenic, mercury, copper, and cadmium, affecting their bioavailability and toxicity [11,12,13,14]. The coating of HA on the surface of magnetite (Fe_3_O_4_) nanoparticles has demonstrated the strong complexation capability of HA with a variety of cations and anions [15,16,17,18,19,20,21,22]. The thin layer of HA coating or film of the nanoparticles can inhibit the auto-oxidation and agglomeration of the bare magnetite nanoparticles (MNP) while preserving the magnetic properties of the iron oxide core critical for post treatment separation of the pollutant-laden nanoparticles [15]. Although the adsorptive properties of HA-MNP have been investigated for the remediation of different environmental contaminants [15,16,17,18,19,20,21,22], their photochemical properties to treat water are largely unexplored [23].

The chromophoric functionality of HA or NOM such as aromatic ketones and aldehydes, quinones, and phenolic compounds can act as a photosensitizer to generate excited triplet states of HA (^3^HA*) and different reactive oxygen species (ROS) [24,25,26,27]. Photoexcitation of HA is regarded as a complex process owing to the diverse and undefined chemical composition of HA. Some major photosensitized reactions of HA are summarized in Figure 1.

The ground state chromophores contained within HA are promoted to their excited states upon the absorption of a photon. The initial excited state can undergo a number of processes, including charge separation with the formation of hydrated electron (e^−^_aq_) and an HA radical cation (HA^●+^) [28] or generation of a singlet excited state (^1^HA*). The molecular oxygen from the surrounding environment can scavenge the hydrated electron to produce superoxide anion radical (O_2_^●−^) [24]. ^1^HA* is relatively short-lived, can undergo fluorescence or internal conversion (IC), leading back to the ground state (HA) [27]. ^1^HA* can produce the lowest excited triplet state of HA (^3^HA*) through intersystem crossing [27]. ^3^HA* can undergo phosphorescence or thermally relax to the ground state. Quenching of ^3^HA* by molecular oxygen via an energy transfer leads to the formation of singlet oxygen (^1^O_2_) [24,27,29]. Reactions of ^3^HA* and the substrate can occur in a number of processes such as triplet–triplet energy transfer, electron transfer, and hydrogen atom transfer to form reduced HA (HA^●−^). HA^●−^ can react with molecular oxygen by electron transfer to generate O_2_^●−^, which can undergo disproportionation in water to generate hydrogen peroxide (H_2_O_2_), Equation (1) [29]. Hydrogen peroxide can react with Fe(II) within HA materials [30] via Fenton and Fenton-like reactions to generate the hydroxyl radical (•OH), Equation (2) [31,32]. Although the formation of •OH can occur through photolysis via HA as shown in Equation (3), this pathway is typically not significant [24,27,29,31,32,33].
(1)$$2O_{2}^{•-}+2H^{+}\to H_{2}O_{2}+O_{2}$$
(2)$${\mathrm{Fe}}^{2+}+H_{2}O_{2}\to {\mathrm{Fe}}^{3+}+\;•\mathrm{OH}+{\mathrm{OH}}^{-}$$
(3)$$\mathrm{HA}+H_{2}O+\;h\nu \to {\;\mathrm{HA}}^{•-}+\;•\mathrm{OH}+H^{+}$$

The adsorptive properties of HA-MNP for the remediation of As(III) and As(V) have been demonstrated by Rashid et al., showing faster removal of As(V) than As(III) [17]. The present study aims to explore the potential photocatalytic transformation of As(III) to As(V) by HA-MNP as a strategy for arsenic removal. The photochemical characteristics of HA-MNP have been assessed via the detection and quantification of the generated ROS, i.e., •OH, ^1^O_2_, and ^3^HA-MNP*. The ROS responsible for the transformation of As(III) to As(V) in the presence of HA-MNP has been assessed in this study.

## 2. Materials and Methods

### 2.1. Chemicals

Sodium arsenite, sodium arsenate dibasic heptahydrate (≥98%), methanol (Optima^TM^ for HPLC), tert-butanol (*t*-BuOH), acetonitrile (HPLC grade), o-phosphoric acid (85%, ACS grade), formic acid (88%), ammonium hydroxide (29.2%), and ferric chloride hexahydrate (FeCl_3_·6H_2_O, 98.8%) were purchased from Fisher Scientific (Waltham, MA, USA). Ferrous chloride tetrahydrate (FeCl_2_·4H_2_O, ≥99%), humic acid sodium salt (Lot#STBC5468V), coumarin (COU) (≥99%), potassium sorbate (≥99%), and 7-hydroxycoumarin (7-HC) (99%) were obtained from Sigma Aldrich (St. Louis, MO, USA). Furfuryl alcohol (FFA) and the tetrabutylammonium hydroxide (TBAH) (40 wt.%) were purchased from Acros Organic (Geel, Antwerp, Belgium). The 2,4,6-trimethylphenol (98%), was obtained from TCI (Portland, OR, USA). The malonic acid (reagent grade, 99.5%) was purchased from Alfa Aesar (Haverhill, MA, USA). Millipore water (MilliQ water, resistivity~18.0 MΩ cm^−1^ at 25 °C) was used for sample and standard preparation unless otherwise indicated.

### 2.2. Synthesis of Humic Acid-Coated Magnetite Nanoparticles

The absorption of humic acid was determined by an Horiba Aqualog Fluorescence spectroscopy (Piscataway, NJ, USA), showing similar characteristics to the reported spectra for NOM. The humic acid-coated magnetite nanoparticles were prepared following an established co-precipitation method [16,18]. Briefly, 3.1 g of FeCl_2_·4H_2_O and 6.0 g of FeCl_3_·6H_2_O were added to 100 mL of MilliQ water. The suspension mixture was heated in a 250 mL three-neck round bottom flask equipped with a reflux condenser with continual magnetic stirring. Once the temperature reached 90 °C, 10 mL ammonium hydroxide (pH ~ 11) was added. The formation of black magnetite nanoparticles was immediately observed [34]. The humic acid-coated magnetite nanoparticles were formed by adding a 50 mL of 1% humic acid sodium salt solution to the reaction mixture rapidly after adding ammonium hydroxide. The material was aged for another 30 min at 90 ± 5 °C. The solution was cooled in an evaporating dish and washed with MilliQ water several times to remove the free HA. The product was dried in a vacuum oven at room temperature. The dried products were grounded with a pestle to a fine powder and stored in a desiccator until use. 

### 2.3. Characterization of Synthesized Materials

The attenuated total reflectance Fourier transform infrared spectroscopy (ATR-FTIR) spectra of the uncoated and coated nanoparticles were obtained using a PerkinElmer FTIR 100 (Waltham, MA, USA) in the spectral range of 400–4000 cm^−1^ with 16 scans per spectrum. The morphology of the synthesized nanoparticles was investigated using scanning electron microscopy (SEM) JEOL 6330F (Peabody, MA, USA) operating at 25 keV. The zeta potentials of HA-MNP and MNP were determined using a Malvern Zetasizer Nano Z (Malvern, Worcestershire, UK). The recorded zeta potentials of HA-MNP and uncoated MNP at different pH values ranging from 2 to 11 were fitted to a Boltzmann sigmoidal function to obtain the isoelectric points [35]. The zeta potentials of HA-MNP in the presence of As(V) before and after the treatment were also measured. Characterization of the synthesized HA-MNP particles using different analytical techniques were reported in detail by this and other research groups [15,16,17,18,19,20,21,22,23].

### 2.4. Experiments

The photochemical reactor (Southern New England UV company, model RPR-100, Branford, CT, USA) contained 14 phosphor-coated low-pressure mercury lamps of 350 nm and a cooling fan. The light flux was reported as 1.6 × 10^16^ photons/sec/cm^3^ using potassium actinometry [36]. A fused quartz cylinder vessel was used as a reaction vessel (L = 200 mm, ID = 25 mm). An initial concentration of 200 µg L^−1^ (ppb) As(III), pH = 6 ± 0.5, was exposed to a series of 0.1, 0.2, and 1.0 g L^−1^ of HA-MNP to study the photo-oxidation of As(III) by HA-MNP. The As-loaded HA-MNP suspension was purged with a specific gas for 15 min prior to and throughout the photolysis period. A 1.2 mL sample aliquot was taken from the suspension at given time intervals and immediately filtered through a 0.45 µm syringe filter before being subject to analyses. The free As species were analyzed using HPLC-ICP-MS (Perkin Elmer, NexION 2000, Waltham, MA, USA) based on the established procedure [37], with a reverse phase Biobasic C18 column (250 × 4.6 mm, 5 µm), a mobile phase of 3 mM of malonic acid, 5 mM TBAH, and 5% (*v*/*v*) of methanol. The pH of the mobile phase was adjusted to 5.5 with malonic acid or TBAH. The flow rate was 1.0 mL/min, and the injected sample volume was 10 µL.

### 2.5. Roles of Molecular Oxygen and ROS

The photoirradiation of HA can lead to the generation of •OH predominantly through indirect reactions as illustrated in Equations (1) and (2), while direct formation represented by Equation (3) is typically insignificant [24,27,29,31,32,33]. Once the generated hydroxyl radical rapidly reacts with As(III), Equation (4), to form an intermediate •As(IV)(OH)_4_ [38], due to the experimental solution pH = 5–7, •As(IV)(OH)_4_ is deprotonated to •HAs(IV)O_3_^−^ as shown below in Equation (5), which is further oxidized to As(V) by reactions with •OH, molecular oxygen or disproportionation as shown in Equations (6)–(9) [38]:(4)$$\mathrm{As}\left(\mathrm{III}\right){\left(\mathrm{OH}\right)}_{3}+\;•\mathrm{OH}\overset{\mathrm{fast}}{\to }\;•\mathrm{As}\left(\mathrm{IV}\right){\left(\mathrm{OH}\right)}_{4}\;\;k=8.5\;\times \;10^{9}{\;M}^{-1}\cdot s^{-1}$$
(5)$$•\mathrm{As}\left(\mathrm{IV}\right){\left(\mathrm{OH}\right)}_{4}↔\;•\mathrm{HAs}\left(\mathrm{IV}\right)O_{3}^{-}+H^{+}+H_{2}O\;\;\mathrm{pKa}=3.64\;\pm 0.05$$
(6)$$•\mathrm{HAs}\left(\mathrm{IV}\right)O_{3}^{-}+\;•\mathrm{OH}\;\to {\;H}_{2}\mathrm{As}\left(V\right)O_{4}{}^{-}$$
(7)$$•\mathrm{HAs}\left(\mathrm{IV}\right)O_{3}^{-}+O_{2}\to \mathrm{HAs}\left(\mathrm{IV}\right)O_{3}---O_{2}{}^{•\;-}\;\;k\approx \;10^{9}{\;M}^{-1}\cdot s^{-1}$$
(8)$$\mathrm{HAs}\left(\mathrm{IV}\right)O_{3}---O_{2}{}^{•\;-}\overset{H^{+}}{\to }\;\mathrm{As}\left(V\right)+{\mathrm{HO}}_{2}{}^{•}/O_{2}^{•-}\;\;k\;\approx \;10^{10}{\;M}^{-1}\cdot s^{-1}$$
(9)$$2\;•\mathrm{HAs}\left(\mathrm{IV}\right)O_{3}^{-}\overset{H_{2}O/H^{+}}{\to }\mathrm{As}\left(\mathrm{III}\right){\left(\mathrm{OH}\right)}_{3}+H_{2}\mathrm{As}\left(V\right)O_{4}{}^{-}\;\;k=4.5\;\times \;10^{8}{\;M}^{-1}\cdot s^{-1}$$

The superoxide anion radical can directly oxidize As(III) to the intermediate As(IV) species as well (Equation (10)) [39]. The oxidation of As(III) can be induced by hydrogen peroxide; however, the slow reaction rate has ruled out this possibility [40].
(10)$$\mathrm{As}\left(\mathrm{III}\right){\left(\mathrm{OH}\right)}_{3}+{\;O}_{2}^{•-}\overset{H_{2}O/H^{+}}{\to }\;•\mathrm{As}\left(\mathrm{IV}\right){\left(\mathrm{OH}\right)}_{4}+H_{2}O_{2}\;\;k=3.6\;\times \;10^{6}{\;M}^{-1}\cdot s^{-1}$$
(11)$$\mathrm{As}\left(\mathrm{III}\right){\left(\mathrm{OH}\right)}_{3}+{\;H}_{2}O_{2}\overset{H_{2}O/H^{+}}{\to }{\;H}_{2}\mathrm{As}\left(V\right)O_{4}{}^{-}+{\;H}_{3}O^{+}\;\;k=5.5\;\times \;10^{-3}{\;M}^{-1}\cdot s^{-1}$$

A study conducted by Carlos et al., reported the reactive oxygen species (^1^O_2_, H_2_O_2_, and •OH) generated by humic acid-coated magnetite nanoparticles [23]. However, the potential use of HA-MNP has yet to be explored as a photocatalyst for the conversion of toxins [23]. The generation of reactive oxygen species and the triplet excited state of ^3^HA-MNP* was carried out under conditions for photocatalysis of As(III) to As(V) herein.

A series of experiments were carried out to establish the potential of HA-MNP as a photocatalyst to convert As(III) to As(V). Under irradiation with 350 nm light in the presence of HA-MNP and As(III), the generation of As(V) was observed under our experimental conditions. Specific conditions were subsequently employed to assess the roles of different reactive species and the triplet excited state of HA-MNP in the conversion of As(III) during conditions of HA-MNP photocatalysis as detailed below. 

### 2.6. Formation of Hydroxyl Radicals

Under photolysis, HA can generate hydroxyl radicals, represented in Equations (1)–(3), which can subsequently oxidize substrates or react with HA via a scavenging process. To assess the generation and roles of •OH during HA-MNP photocatalysis, coumarin was used to trap hydroxyl radicals. The reaction of the hydroxyl radical with coumarin yields the highly fluorescent 7-hydroxycoumarin with ~6.1% yield as reported by Zhang et al., using 100 µM coumarin during TiO_2_ photocatalysis (Equation (12)) [41]. The formation of 7-HC can be accurately monitored using fluorescence and readily correlated to the production of •OH during HA-MNP photocatalysis. Coumarin, which has the triplet state one electron reduction potential of 1.61 V, can be oxidized by ^3^HA-MNP* [42]. A control experiment under argon purge was run to access the oxidation of coumarin by ^3^HA-MNP* to 7-HC.
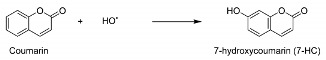
(12)


The concentration of 7-HC was determined after calibration of an Horiba FluoroMax Spectrofluorometer (Piscataway, NJ, USA) using authentic standards of 7-HC with the excitation and emission wavelengths set to 332 and 455 nm, respectively. The formation of 7-HC by •OH reaction follows the pseudo-first-order kinetic model. Formation rates of •OH, R_•OH_ were calculated by dividing the formation rate of 7-HC by the reaction yield. The steady-state concentration of •OH was calculated based on the following equations.
(13)$$\mathrm{COU}+\;•OH\overset{k_{•\mathrm{OH},\mathrm{COU}}}{\to }7-\mathrm{HC}+\;\mathrm{other}\;\mathrm{products}$$
(14)$$\mathrm{COU}+\;•OH\overset{k_{•\mathrm{OH},\mathrm{COU}}}{\to }7-\mathrm{HC}\;Y\;=\;6.1\%\;$$
(15)$$\frac{\;d\left[7-\mathrm{HC}\right]}{\mathrm{dt}}={\;k}_{•\mathrm{OH},\mathrm{COU}}\left[\mathrm{COU}\right]\left[•\mathrm{OH}\right]Y$$
where *K*_•OH_, _COU_ is the reaction rate constant of COU with •OH (5.6 × 10^9^ M^−1^ s^−1^) [43] and *Y* is the trapping efficiency, 6.1% [41].

### 2.7. Formation of Singlet Oxygen

The photoexcitation of HA-NMP produces singlet oxygen at the highest levels among reactive species [23]. The HA-photosensitized generation and partitioning of singlet oxygen occur within the hydrophobic region of chromophoric humic acid solutions [44]. In aqueous media, ^1^O_2_ has a limited diffusion length (~80 nm) with a lifetime of 4 µs [44]. Furfuryl alcohol readily reacts with singlet oxygen, and the disappearance of FFA is commonly used to quantify the production of ^1^O_2_. The generation of ^1^O_2_ was correlated to the loss of FFA following the pseudo-first-order kinetic model. The steady-state concentration was calculated using the following equations [45]:(16)FFA+O12→k1reaction products
(17)d [FFA]dt= −kO12,FFA[FFA][O12]ss
(18)kapp=kO12,FFA[O12]ss

After integration, Equation (17) becomes:(19)$$\ln\frac{\left[FFA\right]}{{\left[FFA\right]}_{0}}=\;-k_{app}t$$
where *k_1O2, FFA_* is the bimolecular reaction rate constant of FFA with ^1^O_2_ (1.2 × 10^8^ M^−1^ s^−1^) [46] and *t* is the time (s). The apparent rate constant (*k_app_*) for the disappearance of FFA is derived from the linear plots of ln([FFA]/[FFA]_0_) against time. To inhibit the •OH-induced degradation of FFA, excess *t*-BuOH was added to the solution to scavenge •OH, thus leaving only singlet oxygen as the predominant species leading to the disappearance of FFA. Under these conditions, the steady-state concentration of ^1^O_2_ ([^1^O_2_]_ss_) was determined by dividing the observed FFA degradation rate by the biomolecular reaction rate constant of FFA with ^1^O_2_. At a low probe concentration, 100 µM, ^1^O_2_ may be deactivated by collision water (*k_d_* = 2.5 × 10^5^ s^−1^) or quenched by HA according to Equation (20) [47].
(20)[O12]ss= r 1O22.5 ×105+ kHA[HA]

The ^1^O_2_ reaction and quenching rate constants, k_HA_, varied from ~10 to 20 mg^−1^ × L × s^−1^ depending on the origins of humic substances [48]. However, a low concentration of HA was used in this study, hence k_HA_[HA] was less than 2.5 × 10^5^ s^−1^. Thus, the formation rate (*r _1O2_*) of ^1^O_2_ can be deduced by the following equation:(21)[O12]ss= r1O22.5 ×105

The concentration of residual FFA was measured using an Agilent Varian ProStar HPLC (Santa Clara, CA, USA) system equipped with a ProStar 410 autosampler and a ProStar 335 photodiode array detector. The analysis was conducted with a reversed-phase C18 column (250 × 4.6 mm Luna, 5 µm), mobile phase of acetonitrile and 31 mM aqueous formic acid (13:87, *v*/*v*), injection volume of 30 µL, and detection wavelength of 220 nm, following a procedure modified from Jaramillo et al. [49]. The flow rate was 1 mL/min, and the analysis time was set at 13 min.

### 2.8. Formation of the Triplet Excited State of HA-MNP

The formation of ^3^HA-MNP* was quantified using the electron transfer probe 2,4,6-trimethylphenol. TMP does not degrade via direct photolysis because of a limited absorption above 300 nm [27]. The degradation of TMP by HA-MNP followed zero-order kinetics in the studied systems, and the observed reaction rate constant *k_TMP_* (s^−1^) was obtained from the equation ${\left[TMP\right]}_{t}=\;{\left[TMP\right]}_{o}-k_{TMP}t$ [50]. The initial transformation rate was calculated as $R_{TMP}=\;k_{TMP}\;\times {\left[TMP\right]}_{o}$. The reaction between TMP and ^3^HA* competes with the quenching of ^3^HA* by triplet ground state O_2_ that leads to the formation of ^1^O_2_ [51]. HA has a spectrum of triplet states, all of which react at different rate constants with TMP, hence the reported second-order rate constants vary among humic substances or DOM isolates, kTMP, H3A−MNP∗ ~ 0.81−10 × 10−9 M−1s−1 [51,52]. The energy required to promote ground-state O_2_ to ^1^O_2_ is 94 kJ mol^−1^; therefore, O_2_ can accept energy from all ^3^HA* moieties [53,54,55]. The generation of ^3^HA-MNP* was correlated to the loss of TMP, and the disappearance of TMP was measured as a function of time. The transformation rate of TMP can be calculated based on the following equations [50]:(22)TMP+H3A−MNP∗ →k1reaction products
(23)RTMP= RH3A−MNP∗× kTMP, H3A−MNP∗ [TMP]0kTMP, H3A−MNP∗ [TMP]0+ kH3A−MNP∗,O2
where RH3A−MNP∗ is the formation rate of [^3^HA-MNP*] and kH3A−MNP∗,O2=5.0 ×105 s−1 is the pseudo-first-order rate constant of ^3^HA-MNP* quenching by triplet ground state O_2_. The argon-saturated environment was applied to minimize the impact of ^3^HA-MNP* quenching by molecular oxygen. The photodegradation experiments of TMP by HA-MNP in the presence and absence of an excess amount of *t*-BuOH were carried out to investigate the impact of other ROS. As a result, the transformation rate of TMP (Equation (23)) can be deduced by the following equation:(24)RTMP ≈ RH3A−MNP∗

The concentration of the TMP was measured using HPLC, with a reverse phase Biobasic C18 column (250 × 4.6 mm, 5 µm), a mobile phase of acetonitrile and 0.1% aqueous phosphoric acid (45:55, *v*/*v*), injection volume of 30 µL, and detection wavelength at 200 nm [56]. The flow rate was 1.25 mL min^−1^ at room temperature, and the analysis time was set at 15 min.

## 3. Results and Discussion

### 3.1. Material Characterization

In ATR-FTIR spectra (Figure 2a), the band at ~600 cm^−1^ denotes the stretching vibrations of Fe-O bonds as seen in both uncoated and coated magnetite nanoparticles. The band at 1559 cm^−1^ in HA and HA-MNP corresponds to the asymmetric −C=O stretching of the carboxylate anion (−COO^−^) of HA [16]; no −C=O stretching was observed in the uncoated magnetite nanoparticle. The energies of the −COO^−^ absorption band depend on several factors such as the electron density, intra- and inter-molecular H bonding, interactions with metal ions, and coupling with other vibrational modes in the molecule. The study reported by Hay and Myneni [57] showed that the structural environment of the carboxyl group affects the energies of the asymmetric stretching vibrations of the −COO^−^ in natural organic molecules. Due to a lower −COO^−^ vibrational energy, 1559 cm^−1^, we conclude that the dominant fraction of the carboxyl groups in our HA and HA-MNP are substituted aromatics. The −COO^−^ symmetric stretching frequency of our HA is 1379 cm^−1^, which is within the range of 1368 and 1382 cm^−1^ for the reported natural organic molecules [57]. The appearance of a strong band in the synthesized HA-MNP at 1400 cm^−1^ can be assigned to the symmetric carboxylate stretching due to the interaction with the iron oxide core or the scissoring of the −CH_2_ group of HA. The broad band at 3200–3600 cm^−1^ in HA is attributed to −OH stretching of alcohol and/or phenol, which largely disappeared in HA-MNP, indicating the complexation between the magnetite core and the humic acid shell. Gu and coworkers [58] proposed two main mechanisms for HA adsorption on the surface of iron oxide: electrostatic attraction and ligand-exchange between the hydroxyl group of iron oxide and carboxyl and hydroxyl groups of humic acid. During the synthesis process (pH ≥ 10), the surface of magnetite particles is hydroxylated, and ligand exchange occurs between the Fe-OH sites on the magnetite surface and the HA [59].

The zeta potential values of bare MNP and HA-MNP in MilliQ water at pH = 7.2 were measured by dynamic light scattering (DLS) as shown in Figure 2b. After coating with humic acid, the zeta potential value of HA-MNP (−41.4 ± 5.46 mV) was more negatively charged compared to the bare MNPs (−23.8 ± 4.64 mV). Under pH = 7.2, ~100% −COOH functional groups of humic acid can be deprotonated to −COO^−^, making HA-MNP surfaces highly negatively charged. Thus, the pH of point of zero charge (pH_pzc_) of HA-MNP decreased to ~3.0 to 4.5, compared with the pH_pzc_ of bare MNP of ~6.0–7.0. The measured pH_pzc_ of HA-MNPs is also consistent with previous studies [15,16]. The zeta potential of HA-MNP in the presence of As(V) during the photolysis was essentially unchanged. The morphological structure and particle sizes of the uncoated and coated MNPs were determined by SEM. Figure 2c,d show that both the humic acid coated and uncoated iron oxide nanoparticles have a spherical shape and a wide range of particle-size distribution. The floc and porosity on the surface texture of the HA-MNP indicate a large surface area.

### 3.2. Formation of ROS and ^3^HA-MNP*

The concentration of 7-HC was quantified from the fluorescence intensity and was used to determine the •OH concentration. A control experiment under argon-saturated conditions showed that the ^3^HA-MNP* oxidized coumarin to form 7-HC in relatively small amounts, shown in Figure 3. After normalization for the control experiment, the formation rate of 7-HC (R_7-HC_) was 3.27 ± 0.21 × 10^−11^ s^−1^, also illustrated in Figure 3. The formation rate of •OH was calculated as 5.4 ± 0.4 × 10^−10^ M s^−1^, which is in the range of the initial formation rate of the hydroxyl radical generated by different humic acid-coated iron oxide nanoparticles reported by Carlos et al. [23]. The steady-state concentration of the hydroxyl radical in HA-MNP solution was 1.3 ± 0.1 × 10^−14^ M, which is comparable to literature-reported values for DOM [60].

A control experiment under argon purge showed that the concentration of FFA, the singlet oxygen trap, was essentially unchanged throughout the irradiation study. Thus, the loss of FFA under oxygen saturation was correlated to the generation of ^1^O_2_, -d[FFA]/dt = d[^1^O_2_]/dt, with the observed kinetic rate constant of ^1^O_2_ generation, 2.08 ± 0.01 × 10^−5^ s^−1^, obtained from the slope of the graph in Figure 4. The steady-state ^1^O_2_ concentration was found to be 1.8 ± 0.1 × 10^−13^ M, and the formation rate was 4.4 ± 0.1 × 10^−8^ M s^−1^, which is also comparable to the values reported by other researchers for the generation of singlet oxygen by either humic substances, aquatic dissolved organic matter, or sunlit surface water [60,61,62]. Under our experimental conditions, the steady-state concentration of ^1^O_2_ was one order of magnitude higher than •OH, analogous to the generation of •OH and ^1^O_2_ by photoirradiation of dissolved organic matter in surface water [60,63].

The triplet state-induced photodegradation of TMP by ^3^HA-MNP* in the presence of *t*-BuOH is similar to that in the absence of *t*-BuOH. The observed kinetic rate constant of TMP was obtained from the slope of the graph in Figure 5, which was 1.6 ± 0.1 × 10^−4^ s^−1^. The ^3^HA-MNP* formation rate was calculated as 8.0 ± 0.6 × 10^−9^ M s^−1^ based on Equations (23) and (24), which is analogous to the literature-reported values of either humic substances or aquatic dissolved organic matter [60,64].

The lower observed rate of formation of the ^3^HA-MNP* measured by the disappearance of TMP compared to the formation rate of singlet oxygen is rationalized below. The energy required to promote ^3^O_2_ to ^1^O_2_ is 94 kJ mol^−1^; hence, ^3^O_2_ can accept energy from all essential different components with ^3^HA* [53,54,55]. The one-electron oxidation potential of TMP E^o^(ArOH^+•^/ArOH) is 1.22 V, higher than the energy required to form singlet oxygen [42]. The reaction of TMP and the low triplet states with one electron reduction potential moieties (E^o^* < 1.22 V) is thermodynamically unfavorable and thus is not effective at quenching the lowest ^3^HA-MNP* states as molecular oxygen, which results in singlet oxygen formation [42]. Molecular oxygen may also more readily diffuse into the HA film than TMP, may have a higher local concentration and thus be a more effective quencher of ^3^HA-MNP*, resulting in d[^1^O_2_]/dt > d[^3^HA-MNP*]/dt.

### 3.3. Effect of Photo-Oxidation on the Adsorption of As Species by HA-MNP

As shown in Figure 6, adsorption of As(III) was significant for the removal of As(III) from water, suggesting the strong potential of HA-MNP for the treatment of As(III). While the adsorption equilibrium appears to be reached only in the case of the highest HA-MNP concentration, dark control experiments over the range of HA-MNP concentrations were run in parallel with the photochemical experiments to allow direct comparisons. In this study, three concentrations of HA-MNP were employed at a fixed As(III) concentration; thus, the proportionality of adsorption sites to As(III) molecules was varied which can have a pronounced influence on the removal of As(III). At low As(III)/HA-MNP ratios, the number of binding sites to As(III) molecules will be highest, decreasing with HA-MNP concentrations. However, at high As(III)/HA-MNP ratios, the number of binding sites relative to As(III) will be lower, which may lead to residual As(III) in solution that may undergo oxidation from photochemical-generated species. Slightly faster removal of As(III) was observed at lower concentrations of HA-MNP under UV irradiation compared to without irradiation, suggesting the photocatalytic transformation of As(III) to more readily adsorbed As(V) species plays a role in the removal of As(III) at lower ratios of the number of binding sites to As(III) molecules. However, with the increase of HA-MNP concentrations and thus higher HA-MNP to As(III) ratios, the As species removal rate with or without UVA irradiation became similar, suggesting the primary contribution to the decrease in As(III) was due to the adsorptive properties of HA-MNP compared to its photocatalytic role. At high HA-MNP concentration (low As(III)/HA-MNP ratio), the number of binding sites on the humic acid increases, hence there is a capacity for the adsorption of As(III) resulting in fast adsorption [12,14].

To confirm whether •OH was involved in the HA-MNP photocatalytic conversion of As(III), 10 mM of *t*-BuOH was added to quench the •OH generated during the experiments. As shown in Figure 7, the removal of As(III) did not change significantly with the addition of *t*-BuOH, which indicates •OH was insignificant as an oxidant for As(III) under our experimental conditions. Although •OH reacts with As(III) via one electron transfer to form an intermediate As(IV) near the diffusion control limit (Equation (4)), and subsequently, to As(V), low •OH production as shown in the coumarin experiment further confirms that under HA-MNP photolysis, •OH does not play a significant role. Another explanation could be the competition of HA-MNP with As(III) for •OH [65].
(25)$$HA-MNP+\;•OH\to HA-MNP_{ox}$$

As shown in Figure 8, a similar rate of removal of As(III) was observed in the presence and absence of molecular oxygen (argon purge). The results demonstrated that under argon saturation ^1^O_2_ and O_2_^●−^/HO_2_• have minimal effects on the photo-oxidation conversion of As(III) using HA-MNP. While humic substances can photosensitize the formation of ^1^O_2_ and O_2_^●−^, HA can also effectively quench ^1^O_2_ on the order of 10^5^ M × C^−1^ × s^−1^ [48,66]. The binding of HA to the magnetite nanoparticles does not appear to significantly change the ability of HA to deactivate ^1^O_2_ generated by HA-MNP [23]. Furthermore, in a study reported by Buschmann et al., the apparent photo-oxidation rate constants of As(III) using humic substances were similar for D_2_O and H_2_O despite the extended lifetime of ^1^O_2_ in D_2_O [67], which is 13 times longer than in H_2_O. Indicating ^1^O_2_ was not critical in the humic substance-photoinduced conversion of As(III), these authors proposed As(III) was mainly oxidized by a phenoxyl radical generated by dissolved organic matter under their experimental conditions [67].

The removal of As(III) at 0.1 and 0.2 g L^−1^ of HA-MNP was faster when potassium sorbate, the triplet quencher, was added to the solution (Figure 9). Sorbate, which contains an aliphatic backbone, could adsorb onto the HA-MNP via hydrophobic interactions. The carboxylate functionality within sorbate can subsequently complex the As(III) species, leading to the observed faster As(III) removal rates. A few studies using arsenic X-ray absorption spectroscopy [68,69] showed that As(III) forms binary complexes with natural organic matter which is accompanied by the covalent bond formation between As(III) and the aliphatic hydroxylic/phenolic and carboxylic groups of the NOM.

Under argon-saturated conditions, ^3^HA-MNP* cannot be quenched by molecular oxygen. We propose that As(III) and ^3^HA-MNP* undergo an electron transfer to form As(IV) and reduced HA-MNP^●−^, Equation (26), followed by As(IV) disproportionation to form As(III) and As(V) as described above, Equation (9) [70]. To further confirm the role of ^3^HA-MNP*, 10 mM of potassium sorbate was added to the system. Potassium sorbate, a salt of sorbic acid, can undergo energy transfer processes with ^3^HA-MNP*. Sorbate is a widely used triplet state quencher with the triplet energy, E_T_ = 239–247 kJ mol^−1^ [42]. Figure 10a shows the formation of As(V) was partially inhibited by the triplet state quencher, suggesting that the triplet state was one of the important key factors controlling the oxidation of As(III) in our photo-irradiation study. The ^3^HA* comprised both high energy triplets (E_T_ ≥ 250 kJ mol^−1^) and low energy triplets (94 ≤ E_T_ ≤ 250 kJ mol^−1^) [42,54]. The high triplet energy components of ^3^HA* account for 15–53% of the total triplet pool depending on its origins [54]. The sorbate can preferentially quench the highly oxidizing triplets.
(26)As(III)+H3A−MNP∗↔ [As(III)⋯H3A−MNP∗] → As(OH)4+ HA−MNP•−

As shown in Figure 10b, the exposure of As(III) solution to oxygen saturation and HA-MNP in the dark control induced the oxidation of As(III) to As(V). The reduced quinone moieties of HA-MNP reduced O_2_ to H_2_O_2,_ and the formed H_2_O_2_ further reacted with semiquinones (SQ) or with complexed Fe(II) to form •OH and induced the oxidation of As(III) [71]. 

In order to confirm the role of •OH in this experiment, an excess amount of *t*-BuOH was added to the system. However, the formation of As(V) stayed relatively constant, as shown in Figure 10b. The oxidation of As(III) was not significantly affected by the *t*-BuOH, which excludes any major role of •OH under our experimental conditions. This finding agrees with the published work by Hug et al., reporting that •OH is not the dominant oxidant [72,73]. When the argon-saturated system was applied, we observed little to no oxidation of As(III). The Fe(II) can induce dark oxidation of As(III) in the presence of dissolved oxygen, possibly through an intermediate Fe(IV) [72,73]. Under our experimental conditions, the humic acid insulates the iron core [16,74] and prohibits the interaction of Fe(II) in the iron core with As(III). From the As(III) oxidation in the presence and absence of oxygen in the absence of UVA light, we proposed that the quinone moieties of HA-coated magnetite nanoparticles can transfer electrons, resulting in As(III) oxidizing to As(V).

The production of As(V) with and without UV irradiation was significantly enhanced in the absence of molecular oxygen. Under UV irradiation in the absence of oxygen (argon saturated) and the presence of sorbate (triplet quencher), minimal conversion of As(III) to As(V) was observed. Under these specific conditions, singlet oxygen is eliminated and sorbate quenched the triplet excited form of the HA, thus shutting down or intercepting the majority of the photooxidative processes leading to the conversion of As(III) to As(V). The highest conversions of As(III) to As(V) occurred with photo-irradiation under argon or oxygen saturation. We propose that the singlet oxygen and triplet excited state can convert As(III) to As(V); however, under argon saturation (preventing the formation of singlet oxygen), the HA triplet state-mediated oxidation becomes dominant. Without irradiation but under oxygen-saturated conditions, partial conversion of As(III) to As(V) was observed likely due to ambient type oxidation processes. In the absence of oxygen, the conversion was dramatically reduced, suggesting molecular oxygen is critical for ambient (dark) humic acid-mediated conversion of As(III) to As(V).

## 4. Conclusions

HA-MNP was synthesized from readily available environmentally friendly materials and was characterized in this study. The As(III) species was oxidized to As(V) in our study, and the fast removal of As(III) from the aqueous solution was mostly driven by the strong adsorption of As(III) onto the HA-MNP. The removal of As(III) by HA-MNP was ~40–90% over 60 min depending on the amount of HA-MNP at a fixed As(III) concentration. Generation of •OH, ^1^O_2_, and ^3^HA-MNP* was detected and quantified; ^3^HA-MNP* and/or singlet oxygen was found to be responsible for the oxidation of As(III) to As(V) depending on the reaction conditions. Our studies confirm the strong adsorption properties of HA-MNP for As(III) and As(V) and illustrate modest enhancement of As(III) removal from solution with simultaneous UV irradiation in the presence or absence of dissolved oxygen. HA-MNP may offer a possible option for the phototransformation and removal of arsenic from contaminated waters; the economic feasibility will depend on the specific treatment objectives. While UVA appears to enhance the overall removal process of As(III) by HA-MNP, further studies are required to assess the potential of HA-MNP for the photocatalytic degradation of pollutants and toxins through the generation of singlet oxygen and ^3^HA-MNP*.

## Figures and Tables

**Figure 1 nanomaterials-10-01604-f001:**
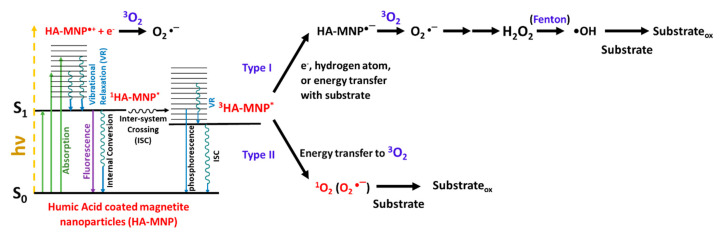
Photogeneration of different HA-MNP excited states and ROS via energy or electron transfer processes.

**Figure 2 nanomaterials-10-01604-f002:**
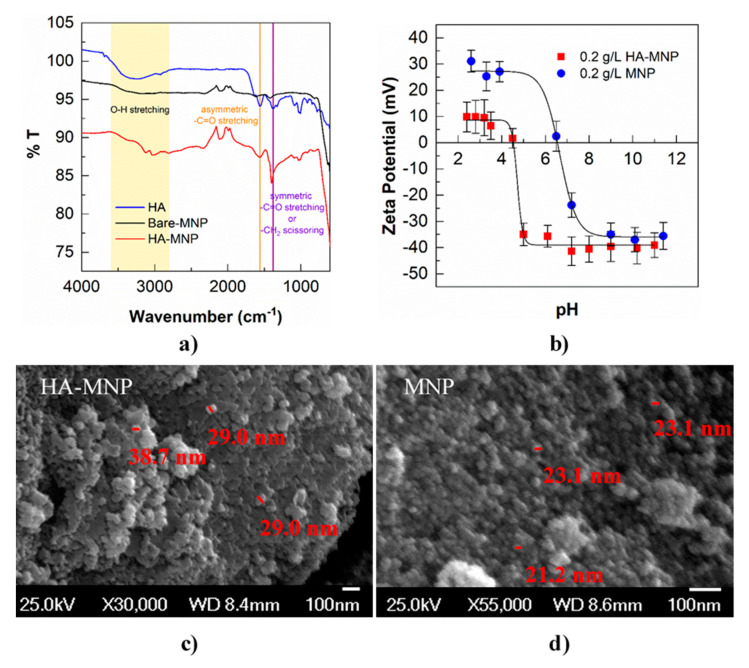
(**a**) FTIR spectra of HA, HA-MNP, and MNP; (**b**) Zeta potential of HA-MNP and MNP; error bars represent the standard deviation of triplicate measurements; (**c**,**d**) Field emission SEM images of HA-MNP and MNP.

**Figure 3 nanomaterials-10-01604-f003:**
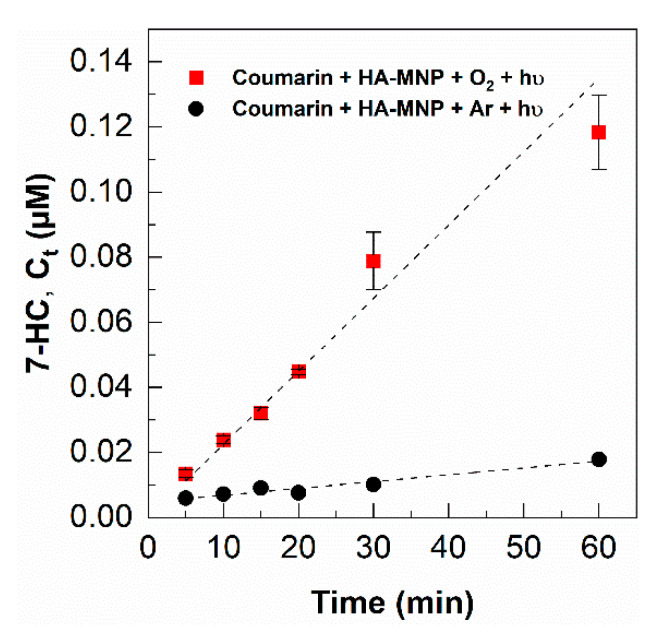
Formation of 7-HC in the presence of UVA and [HA-MNP] = 1.0 g L^−1^; [Cou]_0_ = 125 µM, oxygen-saturated (■) and argon-saturated (●) environments; error bars represent the standard deviation of triplicate measurements.

**Figure 4 nanomaterials-10-01604-f004:**
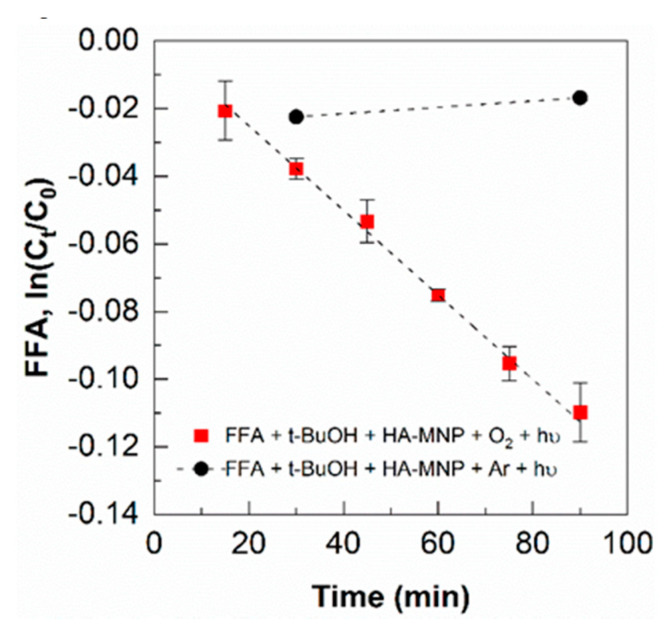
Determination of singlet oxygen generation based on the degradation of FFA, under 350 nm irradiation in the presence of [HA-MNP] = 1.0 g L^−1^; [FFA]_0_ = 100 µM; [*t*-BuOH] = 10 mM, under oxygen-saturated (■) and argon-saturated control experiments (●); error bars represent the standard deviation of triplicate measurements.

**Figure 5 nanomaterials-10-01604-f005:**
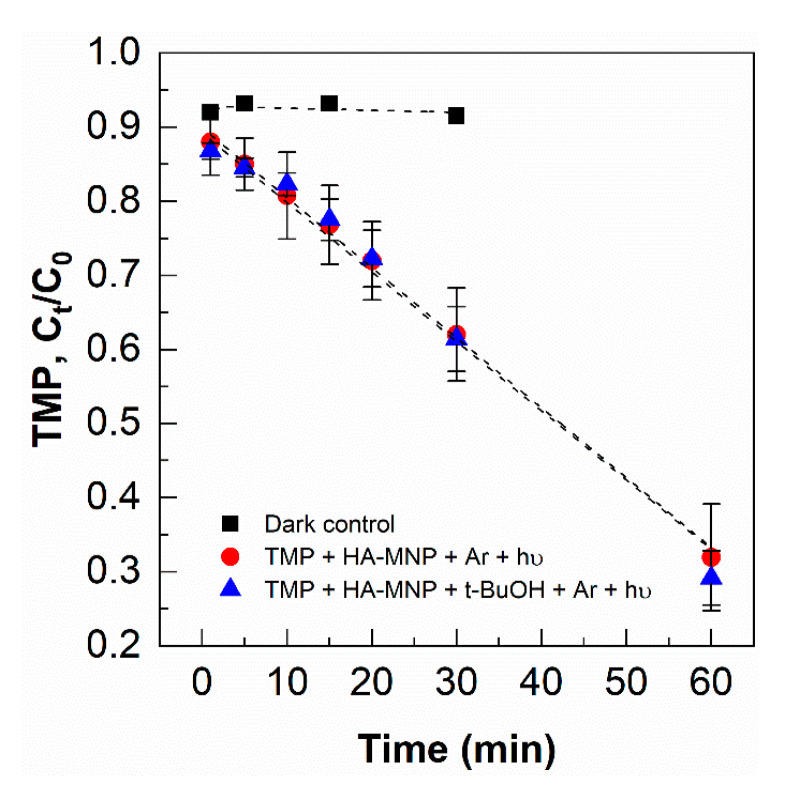
Triplet state-induced photodegradation of TMP in the presence of UVA and [HA-MNP] = 1.0 g L^−1^; [TMP]_0_ = 50 µM, argon saturated; error bars represent the standard deviation of triplicate measurements.

**Figure 6 nanomaterials-10-01604-f006:**
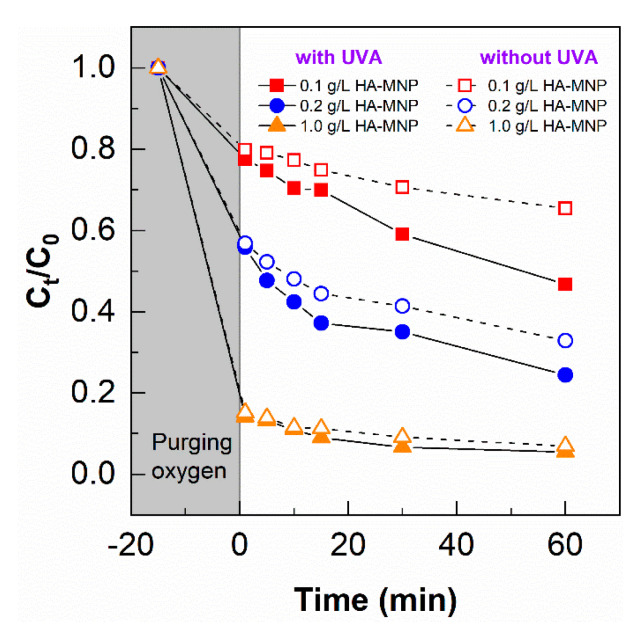
The removal of As(III) in the solution without and with UVA irradiation for 60 min. [HA-MNP] = 0.1, 0.2, and 1.0 g L^−1^; [As(III)]_0_ = 200 µg L^−1^; pH = 6.0 ± 0.5.

**Figure 7 nanomaterials-10-01604-f007:**
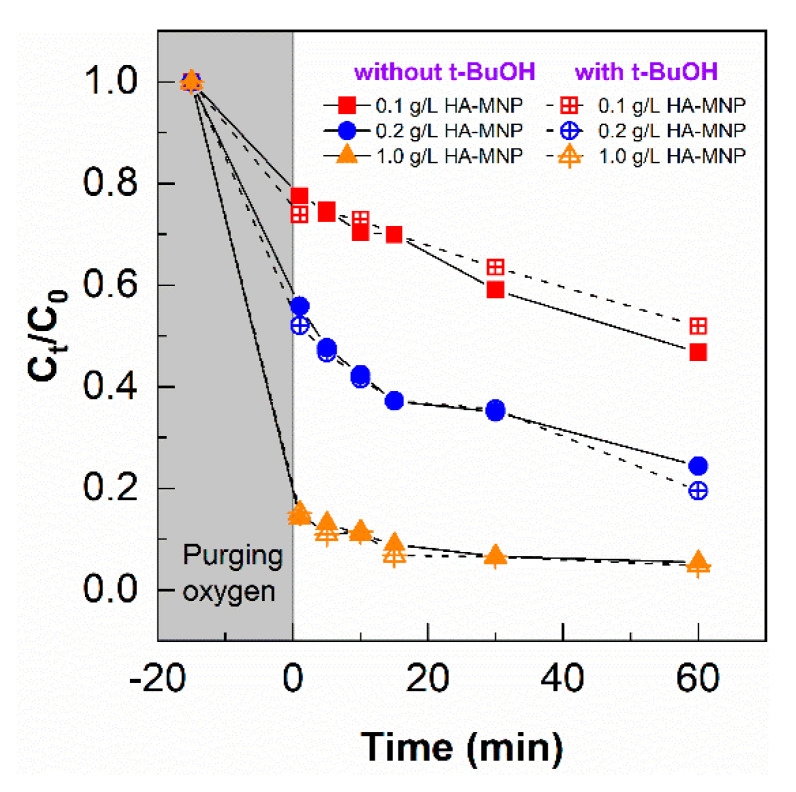
The removal of As(III) in the presence and absence of an •OH quencher during photo-irradiation for 60 min. [HA-MNP] = 0.1, 0.2, and 1.0 g L^−1^; [As(III)]_0_ = 200 µg L^−1^ and [*t*-BuOH] = 10 mM.

**Figure 8 nanomaterials-10-01604-f008:**
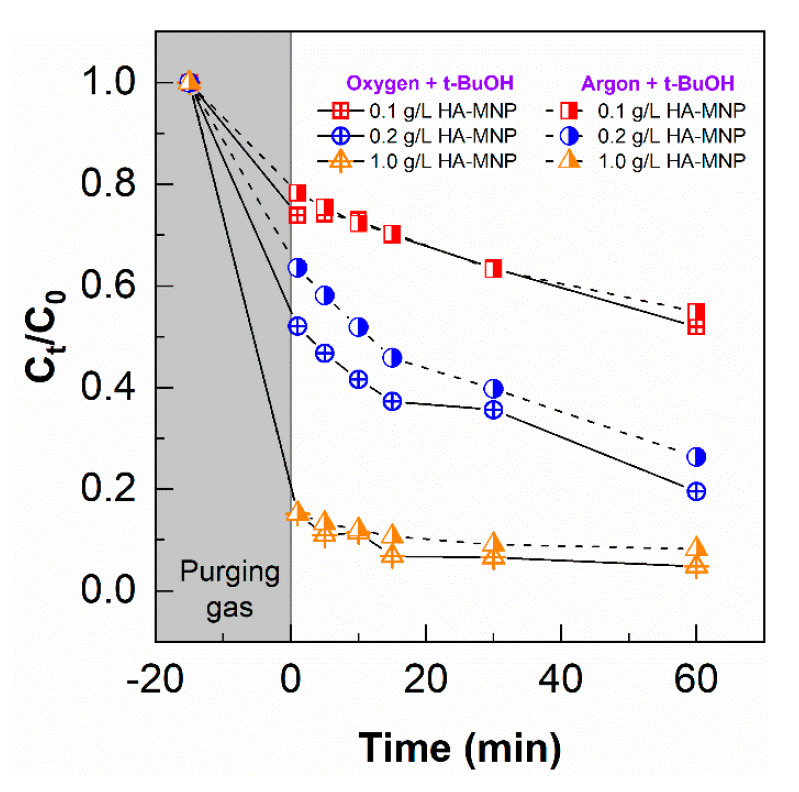
The removal of As(III) in the presence of an •OH quencher and in the presence and absence of molecular oxygen during photo-irradiation with UVA for 60 min. [HA-MNP] = 0.1, 0.2, and 1.0 g L^−1^*;* [As(III)]_0_ = 200 µg L^−1^ and [*t*-BuOH] = 10 mM.

**Figure 9 nanomaterials-10-01604-f009:**
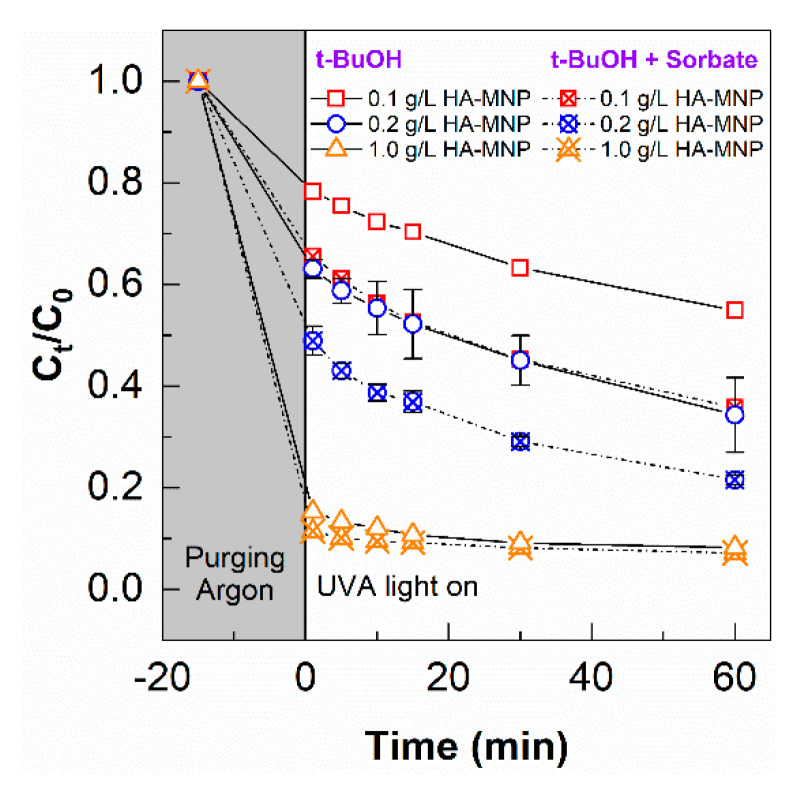
The removal of As(III) in the presence of •OH and ^3^HA-MNP* quenchers and in the absence of molecular oxygen during photo-irradiation for 60 min. [As(III)]_0_ = 200 µg L^−1^, [HA-MNP] = 0.1, 0.2, and 1.0 g L^−1^, [*t*-BuOH] = 10 mM, and [sorbate] = 10 mM; error bars represent the standard deviation of triplicate measurements.

**Figure 10 nanomaterials-10-01604-f010:**
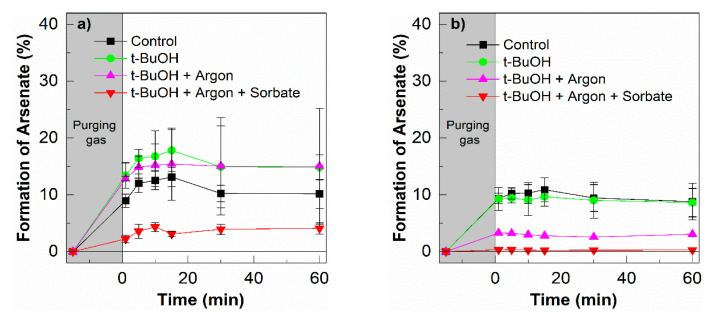
(**a**) Formation of As(V) in the presence and absence of oxygen, or triplet state quencher sorbate during the photo-irradiation with UVA for 60 min. (**b**) without UVA. [As(III)]_0_ = 200 µg L^−1^, [HA-MNP] = 0.2 g L^−1^, [*t*-BuOH] = 10 mM, and [Sorbate] = 10 mM; error bars represent the standard deviation of triplicate measurements.

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
