# Peer review of "Removal of As(III) from Water Using the Adsorptive and Photocatalytic Properties of Humic Acid-Coated Magnetite Nanoparticles"

_nanomaterials, 2020, doi:10.3390/nano10081604_

Round 1

Reviewer 1 Report

The manuscript presents a rather detailed study of photooxidation of As(III) on a surface of humic acid-coated magnetite nanoparticles (HA-MNP). The topic has a strong connection with technologies for arsenic recovery from arsenic-contaminated environments. The main goal was to identify mechanism of As(III) photooxidation on HA-MNP and find key factors influencing on the process. Topic of manuscript is rather new and interesting from the point of view of Water Treatment/Nanotechnology and fits well to the Journal.

However I am not fully agree with main conclusions of the manuscript and have several severe remarks about presentation and discussion of the results. The manuscript also contains numerous technical gaps and typos that make it difficult to understand.  I think that article could be published in the Journal only after major revision.

Major Remarks:

  1. Authors should take care to terms “degradation” and “photodegradation” which are used in the main text and in legends to Figs. 6-9. In fact they measured As(III) removal by both adsorption and (photo)oxidation processes. (Photo)degradation rate could be measured only by monitoring of As(V) concentration, but this information is not presented in the text and figures, for exception of Figure 10.
  2. According to Figure 10 the thermal oxidation of As(III) to As(V) (10%) is twice as effective as photochemical one (about 5%), so why authors pay so many attention to  photocatalytic oxidation of As(III) to As(V) in the studied system? It is not important, especially at high catalyst loading.
  3. In Fig.6 – 10 authors demonstrate the following experimental sequence: bubbling for 20 minutes by argon or oxygen, then irradiation of the mixture by UVA light during additional 60 minutes. May be I missed this moment, by why authors think that after 20 minutes in the dark the adsorption of As(III) at HA-MNP particles comes to equilibrium? In fact all these Figures have to demonstrate dark kinetics in time range of 80 minutes as a control experiment for all studied concentrations of HA-MNP.
  4. Not only initial HA-MNP concentration, but also initial As(III) concentration should influence on adsorption and oxidation of As(III). Authors at least have to discuss the possible influence of initial As(III) concentration on the efficiency of the studied photosystem.
  5. Line 288, Figure 3. Did you have control experiment with irradiation of Coumarin + HA-MNP in argon saturated solution? Please, specify. In principal, oxidation of Coumarin to 7-HC by reaction with 3HA-MNP or other species is possible.
  6. Please, explain why 3HA-MNP* formation rate (7.88 × 10−9 ?‧?−1, Line 302) is lower than formation rate of singlet oxygen (4.34 × 10−8 ?‧?−1, Line 293) though should be opposite trend.
  7. Figure 6. Did you measure the absorption spectra of HA-MNP with concentrations used in the study? It is possible, that at 1 g/L concentration you have very strong absorption of the sample, lower light penetration and, consequently, lower photodegradation of As(III). It could explain the observed effect, presented in Lines 311-314. I recommend to show the absorption spectrum of HA-MNP to ESI or in the main text.
  8. Lines 330 -333. Absence of effect of dissolved oxygen (DO) concentration on degradation of As(III) (Figure 8) also makes doubtful the participation of HA-MNP triplet state in the photooxidation processes, as DO is a common triplet state quencher. If 3HA-MNP* really has a great role in As(III) photooxidation than the removal of DO should accelerate the process but it is not the case.
  9. Lines 345-346 “…therefore, the As(III) species could potentially bound to the diene molecule causing the faster degradation rate”. I don’t see the faster degradation rate of As(III) in this Figure 9 (see also my first remark about using this term). What I see, that in presence of sorbate more As(III) is bound to HA-MNP before irradiation. And upon irradiation the rate of As(III) removal is practically the same in presence and absence of sorbate. So, I suggest reformulating aforesaid statement.
  10. Line 366-367, Figure 10. To support the role of sorbate as a HA-MNP triplet state quencher authors have to do control experiments with “As(V) - HA-MNP – Sorbate” system with and without light and oxygen/argon. I am not sure that As(V) is not simply reduced by sorbate in argon-saturated solution.
  11. Finally I can’t agree with conclusions “HA-MNP materials may offer a unique solution to photochemically transform and removal arsenic from drinking water sources” (Lines 390-391) and “Our results demonstrate HA-MNP may also be useful for the photocatalytic degradation of pollutants and toxins through the generation of singlet oxygen and hydroxyl radicals.” (Lines 392-393). According to Figure 10, 10% of As(III) is converted to As(V) without light and 15% - with UVA light, so UVA irradiation is responsible for only 5% of conversion which is not a very brilliant result. And according to own results of authors both OH and singlet oxygen play a negligible role in As(III) oxidation.

Technical comments:

  1. Some of experimental values in the manuscript have clearly overestimated precision. Please, round all values to their real experimental precision (7.88 × 10-9 to 7.9× 10-9 (Lines 28 and 303), 98 × 10−10 to 8.0× 10−10 (Line 284), and so on throughout the text).
  2. In Figure 1 it will be reasonable to add another channel of O2- radical generation – the reaction of hydrated electron formed by HA photoionization with dissolved oxygen which is described in Lines 68-69.
  3. Line 70, “…to the ground state (0HA)”. Should be (1HA), if the ground state of HA is a singlet one.
  4. I don’t think that HA even in the excited state is so strong oxidant that can oxidize water with formation of OH radical (Reaction 3). I suggest to omit this reaction or to provide some thermodynamical estimates of possibility of such process.
  5. Line 133. “The free As species were analyzed using HPLC-ICP-MS…” it will be great if authors explain briefly how they determine different forms of arsenic, especially As(V), as this is a crucial key to support process of photooxidation.
  6. Reaction (7) and (8) at Lines 144-145. Should be “HAs(V)O3−O2•−” species, not “HAs(IV)O3−O2 • −“.
  7. It is just a remark but the formation of singlet oxygen already proves for the formation of its precursor - 3HA-MNP*. Experiments with 2,4,6-TMP allow to detect only a part of 3HA-MNP* with sufficient red-ox potential if any.
  8. Line 232. “????,2 = 5 × 105 (?−1) is the second-order rate constant of 3HA-MNP*” this is surely apparent first order constant. And, please, correct it to ?3HA-MNP*,2 according to the text on Line 233.
  9. Line 236, should be eq. (23)
  10. What was a source of sorbate used as a triplet state quencher? Is the “diene quencher” in Figure 9 and sodium sorbate (Figure 10) the same compound? Please, specify in “Chemicals”.
  11. Lines 342-343 “when the diene quencher” I think it is better to say “when sorbate, as a triplet state quencher, was added”.
  12. Line 351, legend to Figure 9 “…in the presence and absence of molecular oxygen..” – probably, only in the absence.
  13. Lines 369-377. I don’t understand meaning of this passage. Formation of OH radical in the dark could not be higher than during UVA irradiation. And it was already demonstrated by authors that OH radical play a negligible role in last conditions (Figure 7 and 8).

Reviewer 2 Report

The manuscript entitled ‘Removal of As(III) from water by using the 2 Adsorptive and Photocatalytic Properties of Humic 3 Acid Coated Magnetite Nanoparticles’ presents a complete study that includes the material characterization, the photo-physical properties, the reactive species involved and the performance of the system in the oxidation and removal of As (III). The manuscript is well organized and the information is clearly exposed without enlarging the manuscript artificially. The use of different techniques allows the authors to cover all the processes involved in the removal of As (III).
In the reviewer opinion the manuscript has enough quality and novelty to deserve the publication in this journal. There are only a few questions that could be commented or clarified before the publication:
- Since the authors are employing photocatalytic process, the absorption spectrum of the HA-MNP should be presented.
- The authors employ lamps emitting at 350 nm for its experiments at lab scale. For larger applications, the use of direct solar radiation should be advisable. How could the behavior of the photocatalyst be modified when irradiated with solar radiation?
- Have the authors proven the re-use of the catalyst? Is it possible that HA suffers oxidation and the loss of its initial activity?
- The oxidation from the triplet excited state of the HA-MNP seems to be the main pathway for the oxidation of As (III). However, this mechanism produces the corresponding radical anion HA-MNP·-. As a catalyst, the radical anion has to return to its initial state. Usually, this transformation is attributed to the reaction with oxygen that forms superoxide anion radical. However, the authors show that there is no decrease IN the activity of the catalyst in argon saturated solution. How can the authors explain this fact?

Round 2

Reviewer 1 Report

Authors have done a good job for improving the manuscript and now I feel it worth to be published in the Journal.